# A Study on the O_2_ Plasma Etching Method of Spray-Formed SWCNT Films and Their Utilization as Electrodes for Electrochemical Sensors

**DOI:** 10.3390/s23187812

**Published:** 2023-09-11

**Authors:** Jinkyeong Kim, Ji-Hoon Han, Joon Hyub Kim

**Affiliations:** 1Department of Nanomechatronics Engineering, Pusan University, 2 Busandaehak-ro 63 Beon-gil, Geumjeong-gu, Busan 46241, Republic of Korea; 2KIURI Center for Hydrogen Based Next Generation Mechanical System, Inha University, Incheon 22212, Republic of Korea

**Keywords:** single-walled carbon nanotubes, oxygen plasma etching, electrochemical sensor, electrodes

## Abstract

In this study, we analyzed the morphological changes and molecular structure changes on the surface of single-walled carbon nanotube (SWCNT) films during oxygen plasma (O_2_) etching of SWCNT surfaces formed by the spray method and analyzed their potential use as electrochemical electrodes. For this purpose, a SWCNT film was formed on the surface of a glass substrate using a self-made spray device using SWCNT powder prepared with DCB as a solvent, and SEM, AFM, and XPS analyses were performed as the SWCNT film was O_2_ plasma etched. SEM images and AFM measurements showed that the SWCNT film started etching after about 30 s under 50 W of O_2_ plasma irradiation and was completely etched after about 300 s. XPS analysis showed that as the O_2_ plasma etching of the SWCNT film progressed, the sp^2^ bonds representing the basic components of graphite decreased, the *sp*3 bonds representing defects increased, and the C–O, C=O, and COO peaks increased simultaneously. This result indicates that the SWCNT film was etched by the O_2_ plasma along with the oxygen species. In addition, electrochemical methods were used to verify the damage potential of the remaining SWCNTs after O_2_ plasma etching, including cyclic voltammetry, Randles plots, and EIS measurements. This resulted in a reversible response based on perfect diffusion control in the cyclic voltammetry, and an ideal linear curve in the Randles plot of the peak current versus square root scan rate curve. EIS measurements also confirmed that the charge transfer resistance of the remaining SWCNTs after O_2_ plasma etching is almost the same as before etching. These results indicate that the remaining SWCNTs after O_2_ plasma etching do not lose their unique electrochemical properties and can be utilized as electrodes for biosensors and electrochemical sensors. Our experimental results also indicate that the ionic conductivity enhancement by O_2_ plasma can be achieved additionally.

## 1. Introduction

Carbon nanotubes (CNTs) are tubes composed of carbon atoms with distinctive intrinsic properties. These materials are renowned for their superior mechanical strength, electrical and thermal conductivities, and large surface-area-to-volume ratio [1,2,3,4,5,6,7,8]. The mechanical strength of CNTs surpasses that of steel by several times, and their electrical conductivity is comparable to that of metals such as copper. Their high thermal conductivity makes them desirable for thermal management applications, whereas their vast surface area makes them suitable for various applications such as catalysis and energy storage [9,10,11]. CNTs exist in two forms, single-walled CNTs (SWCNTs) and multi-walled CNTs (MWCNTs), and their properties differ based on their structure characteristics, including diameter and chirality. The chirality of CNTs influences their electronic properties; therefore, SWCNTs can be either metallic or semiconducting depending on their chirality [12,13,14]. These features of CNTs can be utilized by combining them with electrode materials such as electrochemical-based environmental sensors and biosensors utilizing various nanomaterials [15,16].

Numerous researchers have extensively investigated the exceptional attributes of CNTs. Surface modification through etching is one strategy employed to enhance the characteristics of carbon nanotubes for technological applications [17,18,19]. The creation of specific structures, patterns, or shapes that can be utilized for specific applications is a crucial step in the processing of CNTs. For example, etching can regulate the length of the CNTs to create arrays of specific dimensions for electronic applications [20]. By modifying the surface of CNTs, etching can also enhance their adhesion to other materials such as polymers or metals, thereby simplifying the integration of CNTs into composite materials. Etching can also be employed to create precise functional groups on the surfaces of CNTs to amplify their reactivity for particular chemical reactions, including catalytic processes [21]. This is particularly advantageous for applications such as hydrogen storage and energy conversion, for which the catalytic properties of CNTs can be enhanced through etching.

There are various types of CNT etching, including chemical, physical, and plasma, each having its own set of advantages and disadvantages. Chemical etching uses chemicals to modify the surface of CNTs, with the primary disadvantage being possible damage to the structure of the CNTs if it is slow [22]. Moreover, precise control of the etching process can be challenging, leading to inconsistent outcomes. Physical etching employs mechanical or thermal methods to remove materials from the surfaces of the CNTs. The disadvantages of physical etching are that it can also cause significant damage to the CNT structures and has problems being precisely controlled, leading to inconsistent results [23]. Plasma etching uses plasma to modify the surfaces of CNTs, and its disadvantages depend on the specific method employed.

O_2_ plasma etching is a widely adopted technique for etching CNTs owing to its numerous advantages. First, it allows the selective etching of CNTs without causing damage to the underlying substrate material, making it an ideal method for the preparation of CNT-based composites [24]. Second, O_2_ plasma etching achieves high etching rates, facilitating the rapid removal of material from the surface of the CNTs [25]. Third, this is a highly controllable process that can produce specific etching patterns on the CNTs, thereby enabling the preparation of CNTs with tailored shapes and dimensions. Fourth, the surfaces of CNTs can be functionalized by introducing various functional groups such as carboxyl or hydroxyl groups, which can be utilized to modify the surface chemistry of CNTs for diverse applications, including catalysis and energy storage [26]. Furthermore, O_2_ plasma etching is compatible with many existing technologies such as photolithography and electron beam lithography, making it easy to integrate into existing processing flows. In summary, O_2_ plasma etching is a valuable technique for preparing CNTs with specific properties that are crucial for various applications in materials science, electronics, and energy research.

This paper presents a low-cost method for patterning CNTs while preserving their electrical properties, using O_2_ plasma etching to achieve a high etching rate. The experimental results demonstrate that this method can achieve the selective and controlled etching of CNTs with specific patterns, which is necessary for various technological applications.

## 2. Materials and Methods

### 2.1. Materials and Instruments

The SWCNTs used in the experiments were purchased from Hanwha Nanotech Chemical Co. (Seoul, Republic of Korea) and used without further purification. Dichlorobenzene (DCB) at a concentration of 99.0% was employed as a solvent for the SWCNTs. For the selective etching of the SWCNTs, a positive photoresist (GXR-601) from AZ Electronic Materials was used as the passivation layer. The substrate on which the SWCNTs were spray-coated was a borofloat slide glass purchased from iNexus (Seoul, Republic of Korea). The equipment used for the spray coating of SWCNTs was self-made, and the spray rate was adjustable from 0.01 to 10 mL·cm^−2^. For the etching, an O_2_ plasma device from Femto Science (Hwaseong-si, Republic of Korea) was used. The thickness of the SWCNTs was measured using an atomic force microscope (AFM) from Park Systems (Suwon-si, Republic of Korea), which is capable of precise measurements at the nanometer level.

### 2.2. Preparation Process of SWCNTs Films

The experiments to characterize the O_2_ plasma etching were performed as follows. First, a glass slide used as the substrate was ultrasonically cleaned in acetone (99.5%), methanol (99.6%), and deionized (DI) water to remove organic matter. Subsequently, GXR-601, a positive photoresist (PR) that is widely used in semiconductor processing, was spin-coated onto the surface of the glass slide at 3000 rpm. Then, using a fabricated mask, a 1 × 1 cm^2^ area was UV-exposed for 20 s and soft-baked on a hotplate at 95 °C. Finally, the glass slides were patterned with a PR developer so the 1 × 1 cm^2^ area of the slide glass was open, and the rest of the area was left with a positive PR.

In this study, the SWCNTs were formed using the spray method. For this purpose, 3 mg of SWCNTs powder was added to 150 mL of DCB (99.0%) and then ultrasonically dispersed. The dispersed SWCNTs solution was then sprayed onto an open 1 × 1 cm^2^ area of the previously prepared glass, in amounts of 0.01, 0.02, 0.05, 0.1, 0.2, 0.5, 1, 2, 5, and 10 mL·cm^−2^, and dried on a hotplate at 80 °C for 1 h. The SWCNTs formed after drying were strongly bound to the glass slide substrate and could not be removed by N_2_ flow. Finally, the remaining SWCNTs were removed using a lift-off process, except for the 1 × 1 cm^2^ area of SWCNTs left open on the slide glass surface. As shown in Figure 1a, the nozzle of the spray device was positioned perpendicular to the glass slide surface and could move along the *x*-, *y*-, and *z*-axes. In this experiment, the *z*-axis of the nozzle was set at 5 cm from the glass slide surface. Figure 1b shows a photograph of the fabricated spray device.

## 3. Results and Discussion

### 3.1. Thickness and Resistance of Sprayed SWCNTs

Scanning electron microscopy (SEM) images, thickness, and electrical resistivity were measured for each SWCNT layer formed using homemade spraying equipment at spray volumes of 0.01, 0.02, 0.05, 0.1, 0.2, 0.5, 1, 2, 5, and 10 mL·cm^−2^. As shown in Figure 2, the morphology of the SWCNTs became clearly visible as the spray volume gradually increased. The diameters of SWCNTs were of approximately 1–2 nm at relatively low spray volumes (0.01–0.2 mL·cm^−2^) and of approximately 2–3 nm at high spray volumes (0.5–10 mL·cm^−2^). A net-like layered structure was observed to form at a spray volume of 0.1 mL·cm^−2^.

Atomic force microscopy (AFM) was used to measure the thickness of the formed SWCNT layers by scanning the tip of the AFM in the direction of SWCNT film formation on the surface of the glass, as shown in Figure 3a. The 3D image of the SWCNT layer obtained by AFM showed a clear three-dimensional view of the net-like layered structure, confirmed by the SEM image in Figure 2. Figure 3b shows the thickness of the SWCNT layer formed at a spray volume of 1 mL·cm^−2^, representative of the SWCNTs solution spray volumes used, with an average thickness of approximately 81 nm.

Figure 3c shows the thicknesses and electrical resistivities of the SWCNT films at the same spray volumes shown in Figure 2. Unfortunately, we were unable to measure the thicknesses of the films at spray volumes of 0.01, 0.02, and 0.05 mL·cm^−2^. Considering that the thickness of SWCNT films at a spray volume of 0.1 mL·cm^−2^ is of approximately 8 nm, it can be concluded that it is difficult to measure the thickness at spray volumes below 0.01 mL·cm^−2^. When the thickness of SWCNT films formed at various spray volumes is plotted on a logarithmic scale, as shown in Figure 3c, it can be observed that the thickness increases exponentially; particularly, the thickness of films formed at the highest spray volume of 10 mL·cm^−2^ is of approximately 498 nm. Furthermore, the electrical resistivity of the SWCNT films decreases with increasing thickness. Table 1 lists the quantitative results shown in Figure 3c for easier analysis of the electrical resistivity and thickness as functions of the spray volume.

### 3.2. Etch Rate as a Function of O_2_ Plasma Etch Times

Figure 4a shows the thicknesses of the SWCNT film and the calculated etch rate as a function of etching time at 50 W. At the beginning of the O_2_ plasma etching, at approximately 30 s, the etch rate was high; however, it decreased with the thickness of the SWCNT film. Figure 4b shows SEM images of the surfaces of SWCNT films formed at a spray volume of 1 mL·cm^−2^, varying the exposure time to O_2_ plasma. To verify the etching of the SWCNTs by O_2_ plasma, the lower half parts were protected by GXR-601 photoresist. During O_2_ plasma etching, the surface morphology of the SWCNT film initially remained almost unchanged. However, a gradual etching process started at 30 s, and most of the SWCNTs were removed after 180 s.

### 3.3. XPS Characterization of SWCNTs by O_2_ Plasma Etching

After the O_2_ plasma treatment, changes in the chemical composition of the SWCNT films’ surface were determined using Raman spectroscopy and X-ray photoelectron spectroscopy (XPS). In previous studies, the Raman spectra of CNTs before and after O_2_ plasma treatment showed D and G peaks at approximately 1324 cm^−1^ and 1595 cm^−1^, respectively [27,28]. The G peak is typically observed in materials composed of graphite, whereas the D peak is more intense in materials with defective structures [27]. This is because the oxygen ions in the high-energy state of the O_2_ plasma can react with carbon atoms to break π-bonds and create defects [29,30]. In the process of repairing these defects, some carbon atoms rehybridize from *sp*2 to *sp*3 to form new bonds with oxygen or other carbon atoms [31]. Therefore, the D peak appears stronger with longer O_2_ plasma treatment times. Figure 5 shows the XPS results of the SWCNT films before and after O_2_ plasma etching. Figure 5a shows that the spectrum of the C1s peak can be divided into five according to its composition. Two peaks corresponding to *sp*2 and *sp*3 appear at 285.1 ± 0.5 and 284.35 ± 0.5 eV and indicate the presence of carbon atoms that are not bonded to oxygen atoms, which are the G-peak and D-peak, respectively, in the Raman spectrum. The remaining three peaks at 286.37 ± 0.5, 287.21 ± 0.5, and 289.04 ± 0.5 eV represent carbon atoms bonded to at least one oxygen atom, C–O, C=O, and COO, respectively. As shown in Figure 5a, with increasing time of O_2_ plasma etching, the *sp*2 peak, which represents the planar structure forming the basic hexagonal lattice of graphite, increasingly re-hybridizes to *sp*3, and the peak of the *sp*3 peak tends to decrease. In addition, C–O, C=O, and COO tended to increase as the O_2_ plasma etching time increased, and the overall CO_x_/C ratio also increased. Therefore, based on the XPS results, it can be concluded that the O_2_ plasma directly reacted with the SWCNTs and oxygen species to produce etching. The XPS results of the SWCNTs as a function of the O_2_ plasma power and etching time are summarized in Table 2.

Apart from the etching of SWCNTs by O_2_ plasma, it has been shown that the DCB used as a solvent to form SWCNT films generates monopolymers on the surface of SWCNTs during the ultrasonic dispersion of SWCNT powder. In other words, when the SWCNTs are ultrasonically dispersed in DCB and the pressure and temperature can reach 100 MPa and 5000 K, respectively, the radicals from the decomposition of DCB can attach to the SWCNTs. Another study reported that during the ultrasonic dispersion of SWCNTs in DCB, the C–Cl bond was of approximately 200.1 eV. As confirmed in Figure 5b, the etching of SWCNTs with O_2_ plasma could additionally remove these C–Cl bonds (the percentage of organic Cl was reduced to 3.14%, 1.69%, and 1.37% when the time of O_2_ plasma etching on the SWCNTs was 0, 30, and 60 s, respectively).

### 3.4. Quality Control Evaluation of SWCNTs with and without O_2_ Plasma Etching

The experiments so far have shown experimental results for the etching of SWCNTs by O_2_ plasma. Most importantly, however, there should be no damage to the remaining SWCNTs after the etching and patterning of SWCNTs with O_2_ plasma. As a final experiment, we compared the electrochemical properties of SWCNTs with and without O_2_ plasma etching. Electrodes of screen-printed SWCNTs and electrodes of O_2_-plasma-etched SWCNTs formed by the method previously described in Section 2.2 were prepared to have a diameter of 1 mm each. Each of the prepared SWCNT electrodes was used as a working electrode, and Ag/AgCl and Pt wires were used as reference and auxiliary electrodes, respectively, to measure the cyclic voltammetry (CV) as a function of scan rate in a 3 M KCl solution containing 10 mM potassium ferricyanide.

As shown in Figure 6a,b, the cyclic voltammograms of the SWCNT-based electrode exhibit a perfect diffusion-controlled reversible response with and without O_2_ plasma etching. The observed peak current, $i_{p}$, for a reversible electron transfer can be determined by the Randles–Sevcik equation:$$i_{p}=\left(2.69\times 10^{5}\right)n^{3/2}AD_{o}^{1/2}C_{o}v^{1/2}$$
where n is the charge transfer number, A is the surface area of the working electrode (cm^2^), $D_{o}$ is the diffusion coefficient (7.6 × 10^−6^ cm^2^/s), $C_{o}$ is the bulk concentration of redox species (mol/L), and v is the scan rate (V/s). A plot of the peak current, $i_{p}$, as a function of $v^{1/2}$ is linear for a redox active species under diffusion control. This linear dependency indicates that a mass transport process has occurred in the oxidation process because of diffusion and can be used on a diagnostic indicator for the redox system being characterized. The Randles plot in Figure 6c indicates that the peak current versus square root scan rate curve is linear. This means that even after O_2_ plasma etching, SWCNTs do not lose any of their intrinsic properties and can be utilized as electrochemical electrodes. Figure 6d shows the Nyquist plots obtained from EIS measurements of SWCNT electrodes with and without O_2_ plasma etching. The charge transfer resistance (R_ct_) in the high frequency region is shown to be lower for the O_2_-plasma-etched electrodes. This is likely due to the fact that the SWCNTs protected by the photoresist were affected by the O_2_ plasma, which enhanced their ion transfer properties. These results show that the electrochemical properties of photoresist-protected SWCNTs are not reduced by O_2_ plasma etching.

## 4. Conclusions

In this paper, among the various etching methods for SWCNTs, a patterning method that does not change the electrochemical properties using O_2_ plasma was studied. The etching of SWCNTs using O_2_ plasma has the advantage of being able to easily and quickly remove only the desired area without damaging the substrate, along with simple photolithography. To verify this, in this study, SWCNT films were formed using a self-made solution-spraying device, and the morphological and molecular structure changes of the surface due to O_2_ plasma etching were analyzed using AFM, SEM, and XPS, respectively. When the SWCNT film surface was treated with 50 W of O_2_ plasma, the etching process started at about 30 s, and after 300 s, almost all of the SWCNT film was removed, which was confirmed by AFM and SEM. To determine the molecular structure changes of SWCNTs during the etching process of SWCNTs by O_2_ plasma, samples after 30 and 60 s of etching were analyzed by XPS. The results showed that the C–O, C=O, and COO peaks increased simultaneously as the O_2_ plasma etching progressed, indicating that the SWCNTs were etched by O_2_ plasma by reacting with oxygen species. To confirm the changes in the electrochemical properties of SWCNTs before and after O_2_ plasma etching, cyclic voltammetry, EIS measurements, and analysis were performed using SWCNTs as working electrodes in a 3 M KCl solution containing 10 mM potassium ferricyanide. As a result, the SWCNT film electrode remaining after O_2_ plasma etching showed a reversible oxidation/reduction reaction based on perfect diffusion control, and the Randles plot of the peak current versus square root scan rate curve showed linearity. In addition, EIS measurements showed that the charge transfer resistance of SWCNTs before and after O_2_ plasma etching was the same, confirming that there is no problem in utilizing O_2_-plasma-etched SWCNT electrodes as electrochemical sensors. These results indicate that SWCNTs can be utilized for electrochemical-based bio, environmental, and chemical sensors.

## Figures and Tables

**Figure 1 sensors-23-07812-f001:**
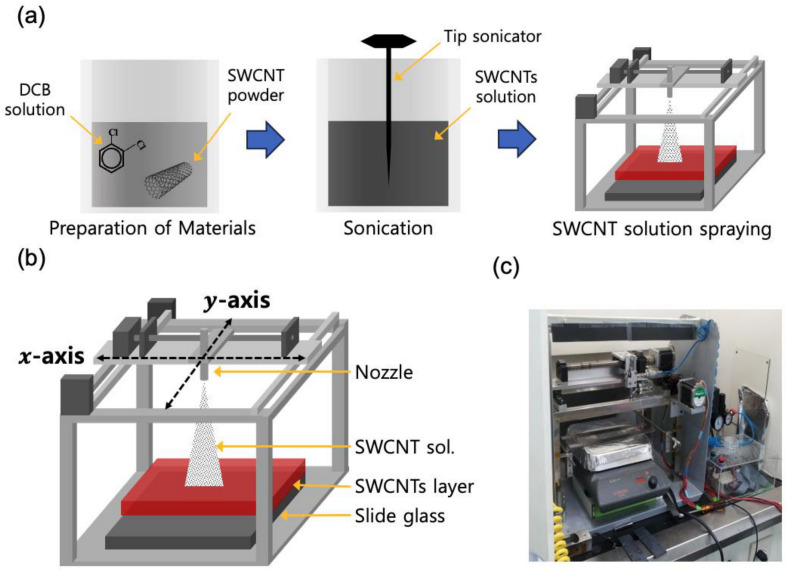
(**a**) Schematic of the SWCNTs solution preparation process. Homemade spray device with spray nozzle movable in *x*-, *y*-, and *z*-axes. (**b**) Schematic diagram and (**c**) a photograph of the spray device.

**Figure 2 sensors-23-07812-f002:**
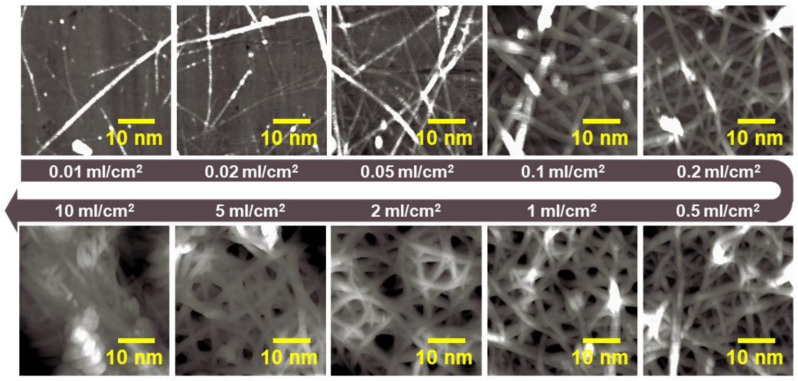
SEM images of SWCNTs formed at different spray volumes (0.01–10 mL·cm^−2^).

**Figure 3 sensors-23-07812-f003:**
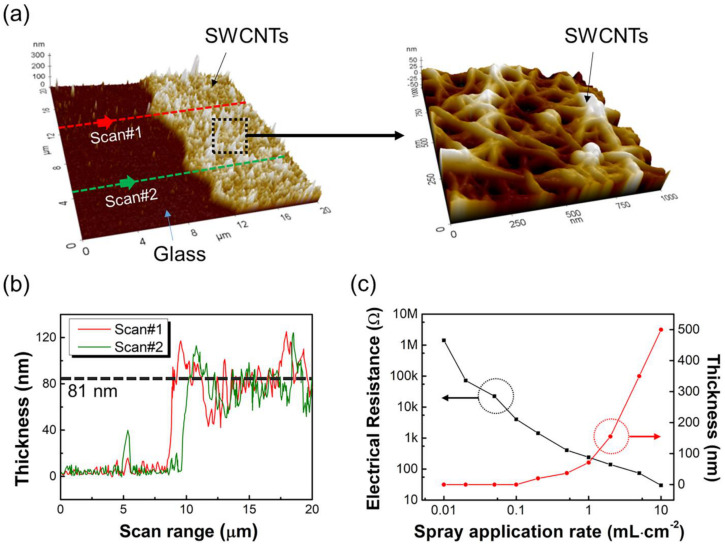
(**a**) Scanning direction of the AFM’s tip for thickness measurement of the SWCNT layer, (**b**) thicknesses of SWCNT films formed at 1 mL·cm^−2^ spray volume, (**c**) thicknesses and electrical resistances of SWCNT films from 0.01 to 10 mL·cm^−2^ spray volume.

**Figure 4 sensors-23-07812-f004:**
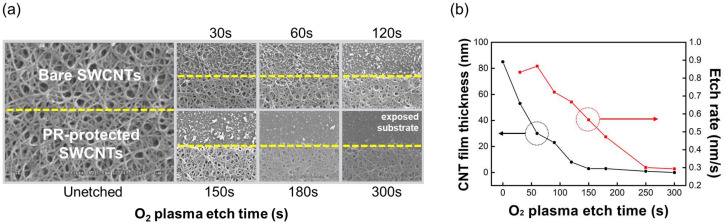
(**a**) Thicknesses and etch rates of SWCNT films depending on the time of O_2_ plasma etching at 50 W, and (**b**) SEM images of the SWCNT film.

**Figure 5 sensors-23-07812-f005:**
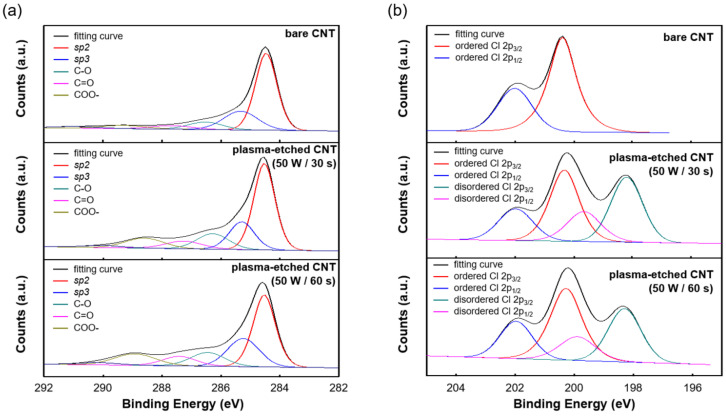
XPS analyses of (**a**) carbon and oxygen species, and (**b**) Cl species of SWCNT films varying etching time in O_2_ plasma at 50 W.

**Figure 6 sensors-23-07812-f006:**
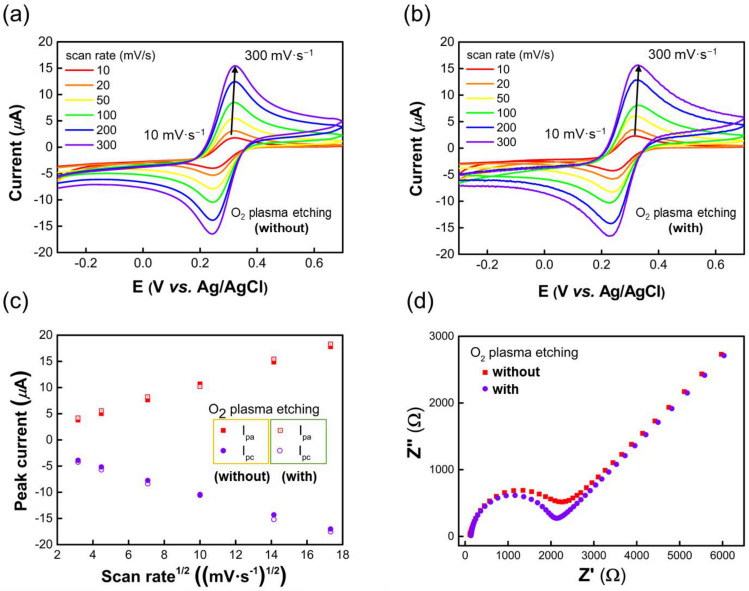
Cyclic voltammograms for SWCNT electrodes in 3 M KCl solution containing 10 mM potassium ferricyanide (**a**) without O_2_ plasma etching and (**b**) with O_2_ plasma etching. (**c**) Randles plot of peak current versus root mean square scan rate, and (**d**) EIS measurement results with and without O_2_ plasma etching.

**Table 1 sensors-23-07812-t001:** Values of CNT films’ thicknesses and electrical resistances depending on the amount of sprayed SWCNT solution.

Spray Volume (mL·cm^−2^)	CNT Film Resistance (Ω)	Thickness (nm)
0.01	20 M	-
0.02	1 M	-
0.05	45 k	-
0.1	20 k	8
0.2	5 k	20
0.5	1.8 k	37
1	800	85
2	500	155
5	160	346
10	70	498

**Table 2 sensors-23-07812-t002:** XPS results for elements and composition of SWCNTs following O_2_ plasma etching.

Comparison Samples	*sp*2	*sp*3	C–O	C=O	COO–	CO_x_/C
bare CNT	69.04%	16.79%	6.70%	3.17%	2.89%	12.76%
plasma-etched CNT (50 W/20 s)	57.85%	18.64%	10.31%	5.14%	6.68%	22.13%
plasma-etched CNT (50 W/20 s)	52.51%	20.64%	9.98%	6.93%	8.31%	25.22%

## Data Availability

Not applicable.

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
