# Peer review of "A Study on the O2 Plasma Etching Method of Spray-Formed SWCNT Films and Their Utilization as Electrodes for Electrochemical Sensors"

_sensors, 2023, doi:10.3390/s23187812_

Round 1

Reviewer 1 Report

Thanks a lot for the opportunity to review the manuscript titled " Scheme 2. Plasma etching behavior of spray-formed SWCNT films and their application as electrodes in electrochemical sensors". In this paper, the authors investigated the O2 plasma etching behavior of spray-formed single-walled carbon nanotube (SWCNT) films on a glass substrate. The results revealed that the remaining SWCNTs, after etching, maintain their unique electrochemical properties, making them promising candidates for biosensors and electrochemical sensors. The subject area of the manuscript is quite interesting, and it would certainly add a scientific contribution to the relevant field. I recommend the publication in "Sensors" after minor revision. The following suggestions are provided for the authors' revising manuscript.

1.      In the ''Abstract'', what was the objective of the study regarding the O2 plasma etching behavior of SWCNT films?

2.      The manuscript needs more thorough language polishing, and it may require professional editing exactly in vocabulary collocation, and the connection between the sentences is not very smooth, especially in the introduction section.

3.      How did the Randle plot analysis demonstrate the electrochemical properties of the SWCNTs after O2 plasma etching?

4.      According to the EIS measurements, how did the charge transfer resistance of the remaining SWCNTs compare to their state before etching?

5.      What implications can be drawn from the experimental results regarding the potential utilization of the remaining SWCNTs as electrodes for biosensors and electrochemical sensors?

6.      What were the outcomes of the cyclic voltammetry analysis, and what did it suggest about the remaining SWCNTs' electrochemical behavior?

7.      Please mention all the materials with their Purity.

8.      Authors are advised to adjust the dimensions of Figure 1b.

9.      Please correct the unit of concentration throughout the manuscript; it's ''mL'', not ''ml''. Although, it’s a very basic thing.

10.  Abbreviations should be placed after full names in the first place, appearing throughout the manuscript (abstract and text separately).

11.  All the Figures in this manuscript must be redrawn and arranged according to each other; the Figures' borders should be the same.

12.  In all tables, make the first letter capital. In Figures too!

13.  Please be descriptive for headings in methods as well as results and discussion.

14.  I am not satisfied with the conclusions of this manuscript; please rewrite it to improve the quality of your work.

15.  I did not see any schematic diagram of preparing materials.

16.  To add more value to the introduction and literature part, please cite the latest references. The following papers should be added to REFERENCES: 

DOI:10.1016/j.teac.2021.e00138, DOI:10.1021/acsami.1c07067

The manuscript needs more thorough language polishing, and it may require professional editing exactly in vocabulary collocation, and the connection between the sentences is not very smooth, especially in the introduction section. Please improve the whole manuscript carefully.

Reviewer 2 Report

In this manuscript, authors study the O2 plasma treated (etched) SWCNTs for understanding sp2 and sp3 bonds with related changed C-O, C=O, and COO peaks. Authors reported the correlations about CNT thickness and resistance, CNTs thickness and plasma etching time.

1.     Why the title is “ Scheme 2. Plasma…….” ?

2.     Why DCB used for the solvent ? According to the XPS results, Cl 2p peaks are strong after plasma etching, I don’t understand the influence about the Cl peaks corresponded with the CV results.

3.     Please explain the purpose about the positive photoresist used as the passivation layer.

4.     Authors mentioned that they used SWCNTs, therefore the RBM peaks of before/after O2 plasma etching should be discussed.

5.     About the electrochemical measurements, including the CV and EIS. To be honest, I cant see what’s the major change.

The languague is ok.

Round 2

Reviewer 2 Report

Response is ok for me.

It's ok